# CoT-Seg: Rethinking Segmentation with Chain-of-Thought Reasoning and Self-Correction

## Abstract

Reasoning segmentation is an emerging vision-language task that requires generating a segmentation mask from implicit and often ambiguous language queries, enabled by recent advances in Multimodal Large Language Models (MLLMs). However, state-of-the-art training-based approaches often fail in challenging cases that demand higher-level reasoning or external knowledge. In this work, we introduce **CoT-Seg**, a training-free framework that rethinks reasoning segmentation by combining **chain-of-thought reasoning** with **self-correction**. Instead of fine-tuning, CoT-Seg leverages the inherent reasoning ability of pre-trained MLLMs (*e.g.,* GPT-4o) to decompose queries into meta-instructions, extract fine-grained semantics from images, and identify target objects even under implicit or complex prompts. Crucially, CoT-Seg incorporates a self-correction stage: the model evaluates its own segmentation against the original query and reasoning trace, identifies mismatches, and iteratively refines the mask. This tight integration of reasoning and correction significantly improves reliability and robustness, especially in ambiguous or error-prone cases. Furthermore, we extend CoT-Seg with **retrieval-augmented reasoning**, enabling the system to access external knowledge when the input lacks sufficient information, further enhancing segmentation accuracy. Extensive experiments on ReasonSeg and RefCOCO demonstrate that CoT-Seg consistently outperforms existing baselines while remaining training-free. Our results highlight that combining chain-of-thought reasoning, self-correction, and retrieval augmentation offers a powerful paradigm for advancing reasoning-driven segmentation.

## 1 Introduction

Reasoning segmentation represents a promising step toward vision-language integration, where a system generates a segmentation mask from complex and often implicit language queries. Recent progress has been driven by fine-tuning Multimodal Large Language Models (MLLMs), such as LISA Lai et al. (2023), Seg-Zero Liu et al. (2025a) and Vision Reasoner Liu et al. (2025b), to produce segmentation outputs. Despite their success, these methods struggle with cases that require nuanced reasoning, domain knowledge, or contextual inference which are the major challenges that humans naturally handle.

Consider the examples in Figures 1–4. Locating the first-chair violinist requires knowledge of orchestra seating arrangements, not just visual similarity. Differentiating surfers by posture demands reasoning about dynamic body positions. Selecting the correct gym equipment for bicep training requires understanding functional affordances. Identifying unracked dumbbells requires contextual analysis of their relation to the rack. These examples highlight that stronger reasoning ability, together with mechanisms to evaluate and refine initial predictions, is essential for advancing reasoning segmentation.

In this work, we present **CoT-Seg**, a training-free framework that rethinks reasoning segmentation through the synergy of *chain-of-thought (CoT) reasoning* and *self-correction*. Rather than relying on fine-tuning or additional training, CoT-Seg leverages the inherent reasoning ability of pre-trained MLLMs (e.g., GPT-4o) to decompose queries into meta-instructions, extract detailed semantics, and produce initial segmentation results. Crucially, CoT-Seg introduces a self-correction stage: the model evaluates its own segmentation against the query and reasoning trace, identifies inconsisten-

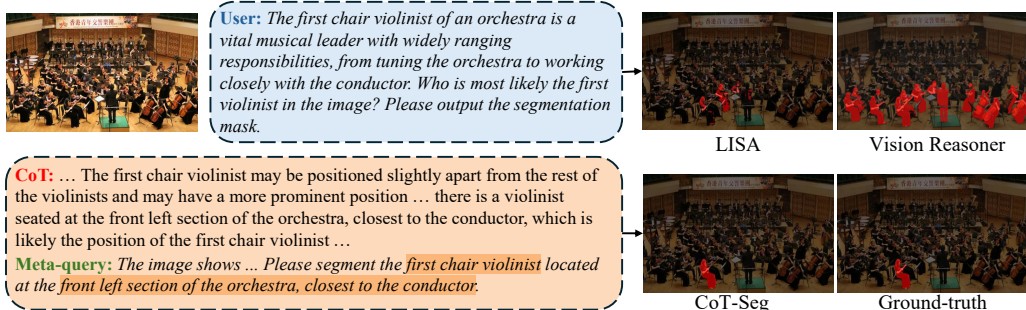

Figure 1: Finding the first violinist (concertmaster) is challenging among similar-looking musicians. CoT-Seg reasons that they sit to the conductor's left and generates a meta-query with relevant *spatial* information, enabling more accurate segmentation than LISA and Vision Reasoner (No self-correction was needed).

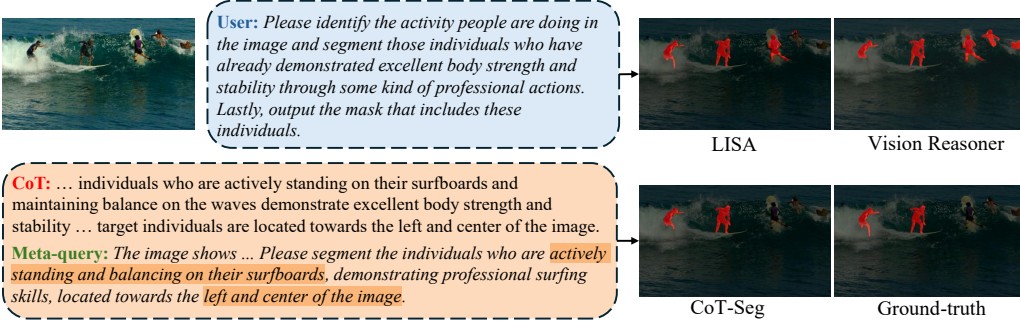

Figure 2: CoT-Seg reasons about the user's query to segment surfers in the correct *pose*, capturing only those who have popped up and are riding waves, unlike LISA and Vision Reasoner (No self-correction was needed).

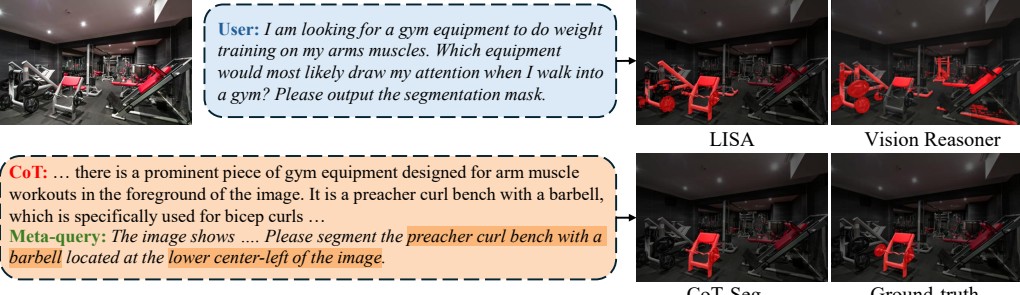

Figure 3: CoT-Seg identifies the gym equipment matching the user's query for biceps, e.g., the preacher's curl, reasoning about its *function* without any training (Self-correction was needed).

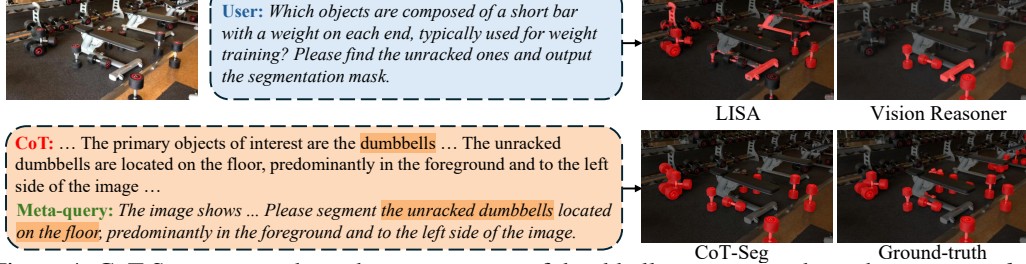

Figure 4: CoT-Seg reasons about the arrangement of dumbbells to segment those that are *unracked*, a more challenging task than simple detection (No self-correction was needed).

cies, and refines the output through automatically generated meta-queries. This feedback loop allows the system not only to think through the segmentation process but also to recognize and repair its own mistakes.

Furthermore, we extend CoT-Seg with *retrieval-augmented reasoning*. When the query and image lack sufficient information, CoT-Seg calls an external agent to retrieve relevant knowledge from the

web, integrating it into the reasoning process. This augmentation further strengthens its ability to tackle ambiguous or knowledge-intensive cases.

Through extensive experiments on ReasonSeg and RefCOCO, we demonstrate that CoT-Seg substantially outperforms existing methods while requiring no additional training. Our results show that integrating CoT reasoning, self-correction, and retrieval augmentation provides a powerful paradigm for advancing reasoning-driven segmentation toward human-level reliability.

## 2 RELATED WORK

**Image Segmentation and Reasoning Segmentation.** Image segmentation has evolved from early graphical-model-based methods, such as Conditional Random Fields (CRFs) Krähenbühl & Koltun (2011); Chen et al. (2017) and region growing Dias & Medeiros (2019), to deep learning approaches that utilize encoder-decoder architectures Badrinarayanan et al. (2017), dilated convolutions Yu & Koltun (2015), pyramid pooling Zhao et al. (2017), and non-local operators Liu et al. (2015). Instance segmentation He et al. (2017); Cheng et al. (2022) and panoptic segmentation Kirillov et al. (2019); Cheng et al. (2020) further pushed the boundary to finer-grained understanding.

The emergence of foundation models for segmentation, especially the Segment Anything Model (SAM) Kirillov et al. (2023), has revolutionized the field. By training on billions of masks and images, SAM enables promptable, zero-shot segmentation with multimodal inputs like points or bounding boxes. Leveraging SAM with Multimodal Large Language Models (MLLMs) has led to a new line of works on reasoning segmentation Lai et al. (2024); Xia et al. (2024); Zhang et al. (2023a); He et al. (2024); Yao et al. (2025). These approaches generate segmentation masks conditioned on implicit or complex textual queries. However, combining MLLMs with SAM directly often fails in challenging scenarios, such as queries requiring domain knowledge, occluded objects, or intricate structures. In contrast, our work shows that *integrating chain-of-thought reasoning and self-correction* can substantially enhance robustness and accuracy in these difficult cases.

**Chain-of-Thought Reasoning in LLMs and MLLMs.** Chain-of-Thought (CoT) reasoning improves reasoning performance in large language models by decomposing complex tasks into intermediate steps Wei et al. (2022); Wang et al. (2022a); Zhang et al. (2023b); Lyu et al. (2023); Kojima et al. (2022). While CoT has been extensively explored in text-only LLMs, its integration into Multimodal LLMs (MLLMs) is more challenging. Existing approaches often rely on fine-tuning MLLMs with multimodal CoT datasets Mondal et al. (2024); Zhang et al. (2023c); Lu et al. (2022) or introducing intermediate representations like graphs Mitra et al. (2024) or code Surís et al. (2023), which limit accessibility and scalability.

Recent works highlight the potential of *test-time CoT reasoning* in pre-trained LLMs Snell et al. (2025) and its applications in visual reasoning Guo et al. (2022); Lian et al. (2023), robotics Hu et al. (2023), and multimodal planning Yao et al. (2025). Inspired by these trends, our framework leverages carefully designed CoT prompts in a *training-free manner*, enabling MLLMs to reason over images and textual queries, evaluate initial segmentation outputs, and self-correct without additional training.

**Self-Correction and Retrieval-Augmented Reasoning.** While CoT provides step-by-step reasoning, errors in initial predictions can propagate if unchecked. Recent studies in reasoning with feedback Zhao et al. (2025); He et al. (2025) demonstrate that self-evaluation and iterative refinement improve accuracy. Our method explicitly incorporates a *self-correction loop* for reasoning segmentation, allowing the model to detect inconsistencies and refine segmentation masks.

Furthermore, retrieval-augmented reasoning Lewis et al. (2021); Komeili et al. (2021) has shown that external knowledge can enhance reasoning when input information is incomplete. CoT-Seg integrates retrieval mechanisms to access relevant knowledge at test time, enabling more robust segmentation under ambiguous or knowledge-intensive queries.

Overall, our work is positioned at the intersection of *reasoning segmentation, CoT-enabled MLLMs, self-correction, and retrieval augmentation*, combining these advances into a unified, training-free framework that achieves state-of-the-art performance in complex vision-language tasks.

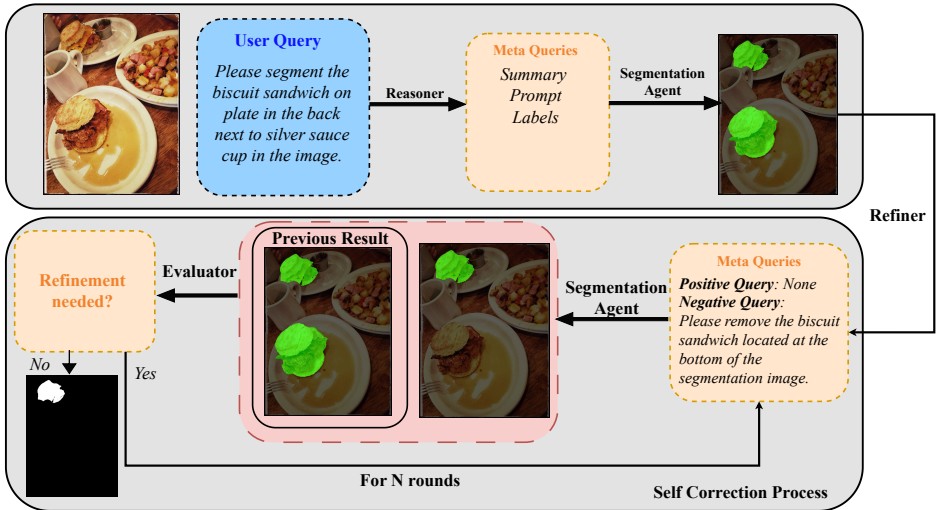

Figure 5: **Overview of CoT-Seg.** The pre-trained MLLM Reasoner generates a chain-of-thought (CoT) over the input image and query, producing an explicit meta-query that translates complex, implicit instructions into clear segmentation guidance. The Segmentation Agent predicts the initial mask, which is then optionally refined by the iterative refinement pipeline. The first-turn mask and original image are examined by the MLLM Evaluator which evaluates the mask and decide if any refinement is necessary. If it does require refinement, then it is passed onto the MLLM Refiner which produces two queries to correct for false positives and negatives. These queries are used inline with the segmentation agent to produce a refined mask for the next iteration of refinement.

## 3 METHOD

Given an image $I \in \mathbb{R}^{3 \times H \times W}$ and a textual query $q$, reasoning segmentation aims to predict a binary mask $\hat{M}$ corresponding to the object(s) referred by $q$. CoT-Seg achieves this by combining chain-of-thought reasoning, self-correction, and optional retrieval-augmented reasoning in a multi-agentic framework. The system consists of three collaborating agents: the MLLM *Reasoner*, the *Segmentation Agent*, and the *Evaluator*.

Fig. 5 gives an overview of CoT-Seg, where the Reasoner analyzes the image and query using a chain-of-thought (CoT) process, generating an explicit meta-query that guides the Segmentation Agent. The Segmentation Agent produces an initial mask using the meta-query and its supported input types, such as text, points, bounding boxes, or scribbles. The Evaluator then analyzes the predicted mask in combination with the original query and image, identifying errors and synthesizing refinement meta-queries for self-correction. This iterative loop allows CoT-Seg to achieve robust zero-shot reasoning segmentation without additional training.

### 3.1 MLLM REASONER

The Reasoner $\mathcal{R}$ performs step-by-step chain-of-thought (CoT) reasoning to identify the target object(s) in the image. To achieve this, $\mathcal{R}$ utilizes a series of *Question Proposers* that generate questions progressively from coarse to fine. Initially, coarse questions capture high-level scene context and object categories. Based on the answers, subsequent proposers generate finer-grained questions to localize the target objects, reasoning over attributes such as position, size, and relationships with other objects. This iterative process continues until sufficient information is collected to precisely identify the target or until it reaches max number of rounds.

Formally, each question-answer pair is generated autoregressively:

$$(Q_k, A_k) = \mathcal{R}(I, q, Q_{<k}, A_{<k}, \text{SegmentorCapabilities}), \quad k = 1, \ldots, n, \tag{1}$$

where SegmentorCapabilities informs the Reasoner which input types the Segmentation Agent supports (e.g., text, points, bounding boxes, scribbles).

After completing all CoT steps, the Reasoner summarizes the collected information into a structured *meta-query* $\tilde{q}_m$, which is compatible with the Segmentation Agent. For non-textual inputs, such as points or scribbles, the meta-query is encoded in a JSON format specifying the input type, coordinates, and spatial attributes:

$$\tilde{q}_m = \mathcal{R}_{\text{summarize}}(\{Q_k, A_k\}_{k=1}^n, \text{SegmentorCapabilities}). \tag{2}$$

This structured meta-query is then passed to the Segmentation Agent to produce the initial mask, and subsequently to the Evaluator for self-correction if necessary. By combining coarse-to-fine question proposing with explicit summarization, the Reasoner ensures precise target localization and effective guidance for zero-shot segmentation.

## 3.2 REASONING SEGMENTATION AGENT

The Segmentation Agent $\mathcal{A}$ predicts masks based on the meta-query $\tilde{q}_m$ and its supported input types. It consists of a frozen vision encoder $E$, a mask decoder $\mathcal{D}$, and a vision-language model $\mathcal{F}$ for multimodal encoding e.g., (Lai et al., 2023; Zou et al., 2023b). The predicted mask is:

$$\hat{M} = \mathcal{A}(I, \tilde{q}_m) = \mathcal{D}(\mathcal{F}(I, \tilde{q}_m), E(I)). \tag{3}$$

By explicitly describing the segmentor's input capabilities, both the Reasoner and Evaluator can adapt their CoT reasoning. If the segmentation agent cannot support a requested input type, the method may fail, highlighting the dependency on the segmentor's flexibility. This design ensures that the meta-query generated by the Reasoner is always compatible with the segmentor.

## 3.3 EVALUATOR AND SELF-CORRECTION

The Evaluator $\mathcal{J}$ assesses the quality of the mask generated by the Segmentation Agent and guides iterative refinement. It receives the original image $I$, the user query $q$, the predicted mask $\hat{M}$, and the SegmentorCapabilities as inputs. The Evaluator performs a chain-of-thought (CoT) reasoning process, similar to the Reasoner, to check whether the mask correctly covers the target objects and respects spatial and semantic constraints.

If refinement is needed, the Evaluator generates two types of meta-queries in a structured JSON format: $\tilde{q}_P$ for false negatives and $\tilde{q}_N$ for false positives. These queries specify the type of correction, spatial coordinates, and other relevant control signals compatible with the Segmentation Agent. Formally, the refinement process is:

$$S = \mathcal{J}_{\text{assess}}(I, \hat{M}, q, \text{SegmentorCapabilities}), \tag{4}$$

$$(\tilde{q}_P, \tilde{q}_N) = \mathcal{J}_{\text{refine}}(I, \hat{M}, q, S, \text{SegmentorCapabilities}), \tag{5}$$

$$s_P = \mathcal{A}(I, \tilde{q}_P), \quad s_N = \mathcal{A}(I, \tilde{q}_N), \tag{6}$$

$$s' = s + \gamma_P \cdot \text{ReLU}(s_P) - \gamma_N \cdot \text{ReLU}(s_N), \quad \hat{M}' = \{(i, j) \mid s'_{i,j} > 0\}. \tag{7}$$

This iterative self-correction loop continues until $S = 0$ (Correct Segmentation) or a maximum number of refinement rounds is reached. By using structured JSON communication, the Evaluator ensures compatibility with diverse Segmentation Agents and input modalities, enabling robust zero-shot segmentation with automated error correction. To ensure that $\hat{M}'$ does not get worse than $\hat{M}$, which may also happen to humans after several refinement turns, $\mathcal{J}$ will make a judgment whether to revert back tothe previous segmentation $\hat{M}$ as the chosen segmentation.

## 3.4 MULTIMODAL INPUT CONTROL

CoT-Seg supports diverse image-based controls in addition to textual queries, including points, bounding boxes, scribbles, and highlighted regions. The Reasoner $\mathcal{R}$ is aware of the Segmentation Agent's capabilities through the SegmentorCapabilities input. For non-textual inputs, it encodes the meta-query in JSON format specifying input type, coordinates, and spatial attributes. This allows both the Reasoner and Evaluator to generate compatible guidance and refinement instructions.

Given an image $I$ and a control image $I_{ann}$, the Reasoner generates step-by-step CoT reasoning to interpret annotated regions and produce a meta-query $\tilde{q}_m$:

$$\tilde{q}_m = \mathcal{R}_{\text{summarize}}(\{Q_k, A_k\}_{k=1}^n, \text{SegmentorCapabilities}, I_{ann}), \tag{8}$$

which is then passed to the Segmentation Agent to produce the mask $\hat{M} = \mathcal{A}(I, \tilde{q}_m)$. The Evaluator can further refine the output via self-correction if necessary, using the same JSON format for multimodal control information.

## 3.5 Retrieval-Augmented Reasoning

In cases where the input image and query do not provide sufficient information, CoT-Seg can augment the Reasoner with an external retrieval step. Specifically, a Retrieval Agent is invoked to search for relevant information from the web or a knowledge database, which is then incorporated into the chain-of-thought reasoning.

For example, consider a query asking to segment a not very famous person in an image. The Reasoner might lack sufficient internal knowledge to identify the individual. The Retrieval Agent searches for information about the person, such as reference images or textual descriptions, and provides these as additional inputs to the Reasoner. The Reasoner then integrates the retrieved knowledge into its CoT reasoning to generate a meta-query, e.g., specifying unique clothing, pose, or contextual cues, which guides the Segmentation Agent to correctly segment the target. Formally, the augmented reasoning can be written as:

$$\tilde{q}_m = \mathcal{R}_{\text{summarize}}(\{Q_k, A_k\}_{k=1}^n, \text{SegmentorCapabilities}, I, I_{retrieved}), \tag{9}$$

where $I_{retrieved}$ contains images or structured data obtained from the retrieval step. This mechanism allows CoT-Seg to handle queries that require external or domain-specific knowledge, extending its reasoning capabilities beyond the information present in the original input.

## 4 Experiments

### 4.1 Experimental Setup

We evaluate CoT-Seg on two recent and widely used benchmarks for reasoning segmentation: ReasonSeg (Lai et al., 2023) and RefCOCO (Kazemzadeh et al., 2014), which covers diverse objects with compositional queries and fine-grained referring expressions. For fair comparison, we use publicly released splits and follow the evaluation protocols of previous works (Lai et al., 2023; Xia et al., 2024; Zhang et al., 2023a). We compare CoT-Seg against state-of-the-art reasoning segmentation methods including LISA (Lai et al., 2023), GSVA (Xia et al., 2024), NextSeg (Zhang et al., 2023a), and MultiSeg (He et al., 2024). As our method is training-free, we emphasize zero-shot evaluation to highlight the effectiveness of inference-time reasoning and self-correction. Performance is measured by Generalized Intersection-over-Union (gIoU) and Complete Intersection-over-Union (cIoU).

### 4.2 Implementation Details

For the Reasoner and Evaluator modules, we use GPT-4o (Hurst et al., 2024) unless otherwise stated, with system prompts tailored for CoT reasoning, summarization, and self-correction. The chain-of-thought reasoning length is adaptively determined by the Reasoner, typically converging within 5–8 steps. The Segmentation Agent is instantiated with Vision Reasoner-7B (Liu et al., 2025b) with SAM-HQ Ke et al. (2023) by default, though we also test compatibility with other SAM-based variants (Kirillov et al., 2023). Structured communication between Reasoner, Evaluator, and Segmentation Agent is implemented in JSON format to handle multimodal control inputs and to ensure capability alignment.

For retrieval-augmented reasoning, we employ a lightweight agent that queries the web using entity names or context keywords extracted by the Reasoner. Retrieved data is passed back as either text descriptions or reference images. To ensure reproducibility, all experiments are run on an NVIDIA 4090 GPU with 24GB memory, although the majority of reasoning computation occurs in the cloud-hosted LLM.

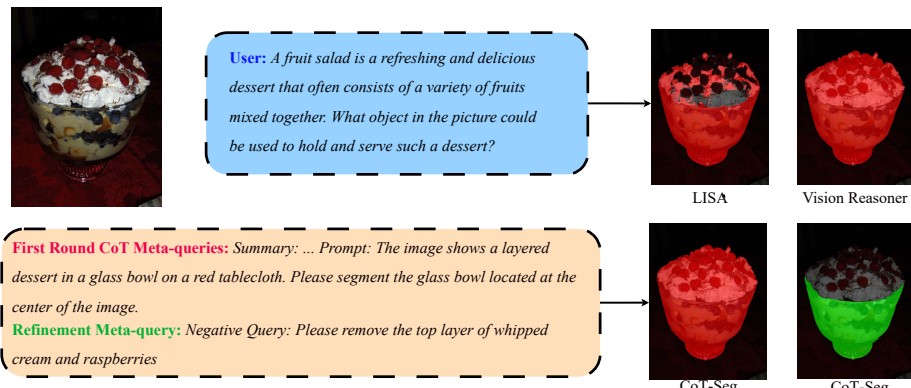

Figure 6: CoT-Seg can identify the actual dessert is not related to the query and self-corrects the segmentation.

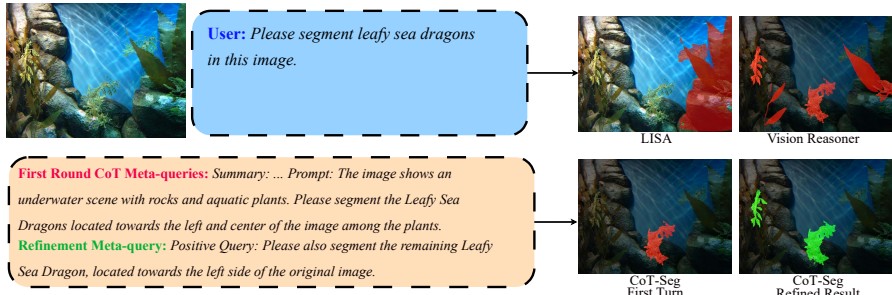

Figure 7: CoT-Seg corrects and finds the missed camouflaged object in self-correction stage.

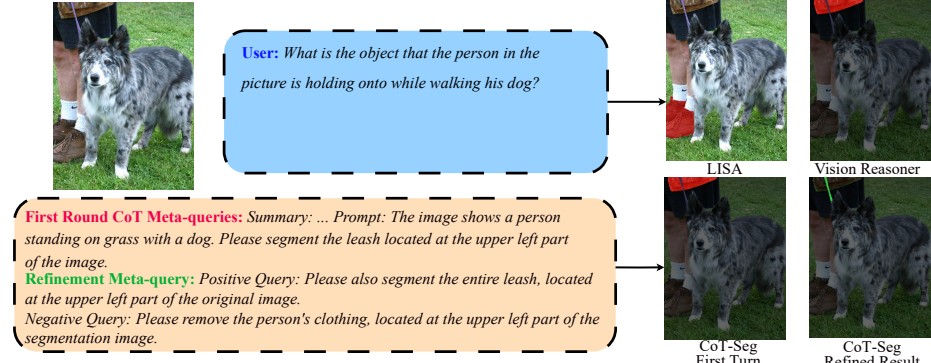

Figure 8: CoT-Seg corrects segmentation by both positive and negative queries.

## 4.3 QUALITATIVE EVALUATION

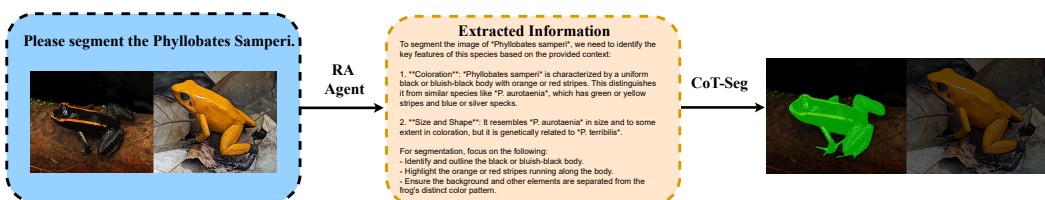

Figure 9: A recently discovered species of frog unrecognizable to GPT-4o. With retrieval augmented (RA) reasoning CoT-Seg was able to segment the frog based on its appearance descriptions from the retrieval agent.

We presented earlier qualitative comparisons in Figures 1–4. More results in Figures 6–10 demonstrate how CoT-Seg progressively reasons about challenging queries and refines initial segmentation masks, demonstrating CoT-Seg's unique capabilities in: 1) resolving implicit queries with multi-step reasoning; 2) correcting masks with fine-grained self-correction (e.g., removing false positives

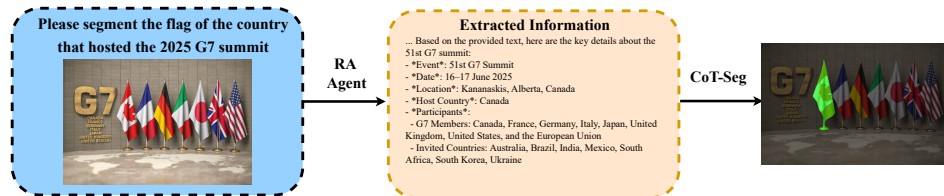

Figure 10: GPT-4o without retrieval augmentation has information up to the 2023 G7 summit, while CoT-Seg can segment the flag of the G7 2025's host country

Table 1: Referring expression segmentation results on RefCOCO Kazemzadeh et al. (2014) dataset. The cIoU metrics of each split are reported. Baselines excerpted from Lai et al. (2024).

| Method | Val. | Test-A | Test-B |
|---|---|---|---|
| MCN Luo et al. (2020) | 62.4 | 64.2 | 59.7 |
| VLT Ding et al. (2021) | 67.5 | 70.5 | 65.2 |
| CRIS Wang et al. (2022b) | 70.5 | 73.2 | 66.1 |
| LAVT Yang et al. (2022) | 72.7 | 75.8 | 68.8 |
| LISA-7B(fine tuned on ReferSeg) Lai et al. (2024) | 74.9 | 79.1 | 72.3 |
| Seg-Zero 7B Liu et al. (2025a) | - | 80.3 | - |
| GSVA-Llama2-13B Xia et al. (2024) | 77.7 | 79.9 | 74.2 |
| GSVA-Llama2-13B (ft) Xia et al. (2024) | **79.2** | **81.7** | **77.1** |
| CoT-Seg | 76.3 | 80.9 | 72.7 |
| CoT-Seg with self-correction | 77.2 | 80.9 | 73.8 |

Table 2: Quantitative evaluation on the test set of *ReasonSeg* (Lai et al., 2023). (ft) means fine-tuning on the train set. † is reproduced with the official released weights with 8-bit quantization.

| Method | ReasonSeg (overall) | |
|---|---|---|
| | gIoU | cIoU |
| OVSeg (Liang et al., 2023) | 26.1 | 20.1 |
| GRES (Liu et al., 2023) | 21.3 | 22.0 |
| X-Decoder (Zou et al., 2023a) | 21.7 | 16.3 |
| SEEM (Zou et al., 2023b) | 24.3 | 18.7 |
| Seg-Zero-7B (Liu et al., 2025a) | 57.5 | 52.0 |
| Vision-Reasoner-7B | 63.6 | - |
| LISA-13B (Lai et al., 2023) | 44.8 | 45.8 |
| LISA-13B-Llama2 (ft)† (Lai et al., 2023) | 50.0 | 51.9 |
| LISA-13B-LLaVA1.5 (ft) | 61.3 | **62.2** |
| CoT-Seg | 66.0 | 58.8 |
| CoT-Seg with self correction | **66.7** | 60.4 |

such as icecream in Figure 6 and recovering missed objects in Figure 7); and 3) retrieval-augmented reasoning for segmenting uncommon entities, such as identifying a new animal species (Figure 9) by integrating retrieved textual and visual cues. These results show that CoT-Seg achieves higher robustness in complex reasoning cases compared to prior methods that rely solely on direct prompt-to-mask predictions.

## 4.4 QUANTITATIVE EVALUATION

Tables 1 and 2 summarize quantitative comparisons across benchmarks. Overall, CoT-Seg achieves SOTA or competitive results in both benchmarks, with the most improvements on ReasonSeg, where high-level reasoning and domain knowledge are essential, while producing improved results after self-correction. Notably, our training-free pipeline achieves higher cIoU than LISA, Seg-Zero and Vision-Reasoner on RefCoco demonstrating the effectiveness of inference-time reasoning and self-correction without requiring additional data or fine-tuning. CoT improves the general score while auto-correction improves the result of hard tasks.

However, our results on RefCoco is outperformed by GSVA Xia et al. (2024) and deviations in general are mainly due to a few cases: i) there exist cases in the dataset where the query is ambiguous which can lead to alternative interpretations of the segmentation task. For example, in Figure 12a (in appendix) prompt is segmenting the pagurian but the ground truth does not include the shell; ii) in Figure 12b the ground truth wrongly includes a leg of the zebra on the right, leading to the segmentation containing five legs. Our result correctly includes the hair at the top of the zebra which the ground truth misses; iii) due to MLLM limitations, CoT-Seg sometimes fail on numerical prompts where object of interest is tightly packed as illustrated in Figure 13a, where counting can be a limiting factor in producing a correct meta-query. Figure 13b, the prompt is *"second from the right,"* the CoT process concludes second from right person wearing purple shirts leading to the wrong result; iv) while further finetuning our framework for a dataset should be able to improve our results, unlike others CoT-Seg is training-free and thus not finetuned to a specific dataset. Our results are dependent on the reasoning segmentor performance as investigated in ablations.

## 4.5 ABLATION STUDIES

**Impact of Self-Correction** Tables 1–2 compare performance with and without the refinement module, showing how iterative refinement improves robustness in ambiguous or cluttered scenes. Qualitative examples are shown in Figures 6–7. Through progressive refinement, CoT-Seg is capable of correcting for the missed object in the first turn results. Overall, CoT-Seg shows SOTA or competitive results with minor deviations discussed in section 4.4.

**Effect of Chain-of-Thought Length** We vary the number of reasoning steps (e.g., 2, 4, 8) to study the tradeoff between reasoning depth and segmentation quality. Table 3 tabulates the results where all of the experiments use a maximum of two rounds of refinements for self-correction running on *RefCoCo Test-A*. The re-

Table 3: CoT length experiment on Test-A of *RefCoco* (Kazemzadeh et al., 2014).

| CoT Length | RefCoco Test-A | |
|---|---|---|
| | gIoU | cIoU |
| CoT-Seg with self-correction using CoT length 2 | 79.9 | 79.4 |
| CoT-Seg with self-correction using CoT length 4 | **80.3** | 79.5 |
| CoT-Seg with self-correction using CoT length 8 | 80.1 | 79.4 |
| CoT-Seg with self-correction using CoT variational length | 80.1 | **80.9** |

sults show that the length of chain of thoughts is not critical to performance, with a length of 4 producing the best score among the tested fixed lengths. Notably, fixed CoT length is outperformed by variational length determined by the MLLM. The results indicates that two reasoning steps usually suffice while overthinking with too many steps may lower the accuracy, with varying lengths depending on the input results in the best accuracy.

**Segmentor Compatibility** In our quantitative experiments, the Segmentation Agent can be different SAM-based or open-vocabulary segmentation backbones. We analyze how their built-in capabilities (text-only prompts, multimodal prompts, or interactive point-based control) affect downstream performance. Table 4 tabulates the results, highlighting the importance of segmentor capability descriptions in guiding Reasoner and Segmentator collaboration.

Table 4: Segmentor experiment without self-correction on Test-A of *RefCoco* (Kazemzadeh et al., 2014).

| Segmentor | RefCoco Test-A | |
|---|---|---|
| | gIoU | cIoU |
| CoT-Seg with Vision-Reasoner-7B + SAM-HQ | **80.1** | **80.9** |
| CoT-Seg with LISA | 77.8 | 79.2 |
| CoT-Seg with GroundedSAM | 51.4 | 61.9 |

**MLLM Agent Variants** We evaluate CoT-Seg with different MLLM backbones, such as GPT-4o, Gemma 3 12b, and Qwen2.5-VL-7B on *RefCoco Test-A* with maximum of 2 rounds of refinement. Table 5 tabulates the results, which reveal how reasoning depth, hallucination tendency, and multimodal grounding influence segmentation quality and stability, showing the tradeoffs between proprietary and open-

Table 5: Different MLLM experiments on Test-A of *RefCoco* (Kazemzadeh et al., 2014).

| CoT with different MLLMs | RefCoco Test-A | |
|---|---|---|
| | gIoU | cIoU |
| CoT-Seg with self-correction using GPT-4o | 80.0 | **80.9** |
| CoT-Seg with self-correction using Gemma 3 | **80.1** | 80.3 |
| CoT-Seg with self-correction using Qwen2.5-VL-7B | 69.2 | 70.3 |

source models in reasoning-driven segmentation. For earlier VL models such as Qwen2.5, when given two segmentations they cannot determine which one is better so they can only fulfill the CoT part and not the auto-correction part of our framework.

**Multimodal Input Control** Our framework can be used for multiple kinds of input including but not limited to bounding box, point, and scribble annotations, demonstrating the flexibility of JSON-based multimodal reasoning and how CoT and auto correction works for all general reasoning strategies, Figure 11. CoT works especially well on improving segmentation based on rough human input, providing important text info for the segmentation agent.

**Retrieval-Augmented Reasoning** We tested scenarios where the input lacks sufficient knowledge (e.g., rare animal species or recent events) and showed improvements when retrieval provides textual or visual cues that complement MLLM knowledge, Figures 9 and 10.

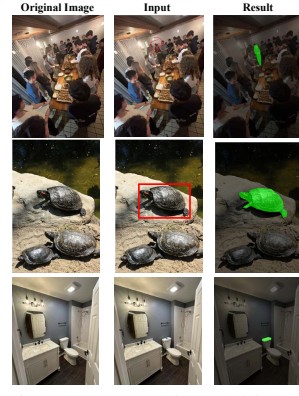

Figure 11: Multimodal inputs

## 5 CONCLUSION

We introduced **CoT-Seg**, a training-free framework that rethinks reasoning segmentation by integrating chain-of-thought reasoning and self-correction with off-the-shelf MLLMs and segmentation agents. Our method enables step-by-step reasoning to synthesize meta-queries, collaborative evaluation for refinement, and retrieval-augmented reasoning for knowledge gaps. Extensive experiments demonstrate that CoT-Seg substantially improves zero-shot segmentation performance across multiple benchmarks. This work highlights the untapped potential of inference-time reasoning and self-correction in bridging vision-language understanding with precise segmentation.

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

# APPENDIX

## A  FURTHER DETAILS

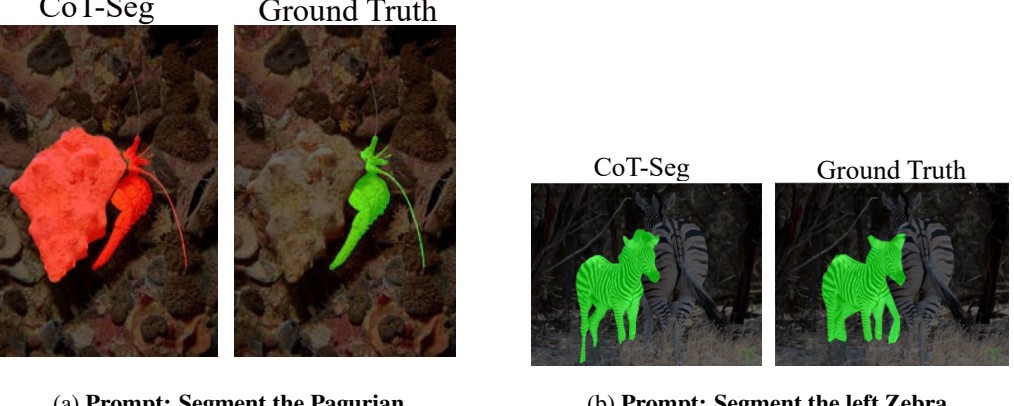

(a) **Prompt: Segment the Pagurian.**              (b) **Prompt: Segment the left Zebra.**

Figure 12

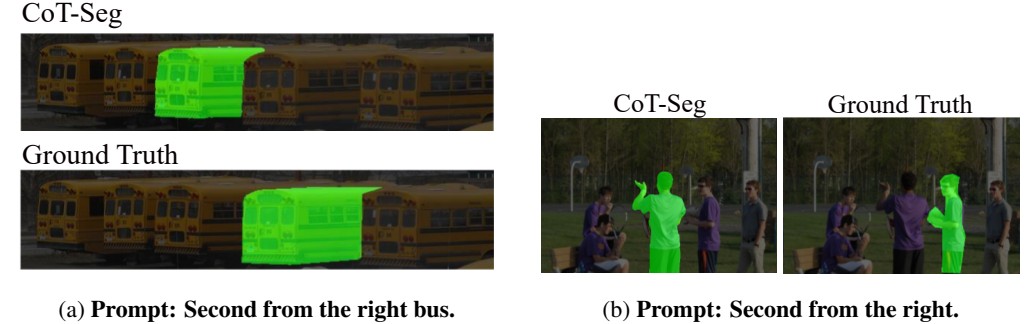

(a) **Prompt: Second from the right bus.**         (b) **Prompt: Second from the right.**

Figure 13

**Figures for 4.4 Quantitative Evaluation**

**CoT First Turn Template**

We use this as a basic description for the LLM to propose questions answer pairs for CoT process, we replace *<QUERY>* with the user query.

*You will serve as an agent for language-based image segmentation model. During each inference, your task is to consider a query and describe a given image with chain of thoughts. You need to provide details to help the segmentation model understand the image better. The target objects may contain multiple layers, be blocked by other object, or be seamlessly embedded in their surroundings. Your description will be later sent to the segmentation as prompt. For example, if given an image, you need to describe what can be seen in the image, the number of objects for each categories, the position of the target object, the structure of the object, the number of layers of the object, etc. The actual description depends on the given image.For the output, you need to follow the format:- Question 1: Answer 1.- Question 2: Answer 2 ..., etc, where each pair of prompt and answer implies the chain of thoughts, i.e., different levels or different part of the image understanding. For example, the first prompt can be related to the overall style or background of the image. Finally, you need to summarize the description based on your generated prompts and answers with strictly with the format: Your summary here ... considering the prompt where the user is looking for ..., the object of interest may be ... Then, based on the summary, you have to generate a pseudo-prompt to query the segmentation model. This pseudo-prompt should contains the information about what is in the image, what to segment, and where the target object is. It must strictly follow the format: - Prompt:*

*The image shows ....Please segment the ... located at ... of the image. Lastly, please generate a list of labels that would be passed to an object detector based on the summary and thinking process that describes the object(s) that fits the user query and should be segmented and in strictly in the format of cat. remote control. television. with a period separating every label and if there is multiple word in a label then separate using space. Do not include other objects not given by the prompt. <QUERY>*

**CoT Self-Correction Process**
*- Reasoning process:*
*1. Original image: The original image shows an underwater scene with various marine life forms and corals. The Pagurian, or hermit crab, is located at the center-right of the image. It is identifiable by its orange color and the shell it occupies.*
*2. Segmentation image: The segmentation image shows an isolated orange Pagurian on a white background. The shell is not included in the segmentation.*
*3. Summary: The segmentation correctly isolates the Pagurian itself but fails to include the shell, which is an integral part of the Pagurian's appearance in the original image. Therefore, the segmentation is incomplete.*
*- Correctness: False*
*- Meta-queries (Output if the correctness is false):*
*Positive: Please also segment the shell of the Pagurian, located at the center-right of the original image.*
*Negative: None*
*- Labels:*
*Positive: shell*
*Negative: None.*

**Full First Turn and Subsequent Self-Correction CoT**
*First Turn*
*- Question 1: What is the overall setting of the image?*
*- Answer 1: The image shows a person standing on grass, with a dog in the foreground.*
*- Question 2: What are the main objects visible in the image?*
*- Answer 2: The main objects visible are a person, a dog, and a leash.*
*- Question 3: What is the person holding?*
*- Answer 3: The person is holding a leash.*
*- Question 4: Where is the object of interest located in the image?*
*- Answer 4: The leash is located in the upper left part of the image, extending from the person's hand to the dog.*
*Summary: The image shows a person standing on grass with a dog. Considering the prompt where the user is looking for the object that the person is holding while walking their dog, the object of interest may be the leash.*
*Prompt: The image shows a person standing on grass with a dog. Please segment the leash located at the upper left part of the image.*
*Self-Correction*
*- Reasoning process:*
*1. Original image: The original image shows a person standing on grass with a dog. There is a leash visible in the upper left part of the image, held by the person.*
*2. Segmentation image: The segmentation image shows a portion of the person's clothing and a small part of the leash in the upper right corner, isolated on a white background.*
*3. Summary: The segmentation result does not correctly reflect the user query. The query asks for the leash to be segmented, but the segmentation image only includes a small part of it and part of the person's clothing. The entire leash should be included, and the person's clothing should be excluded.*
*- Correctness: False*
*- Meta-queries (Output if the correctness is false):*
*1. Positive: Please also segment the entire leash, located at the upper left part of the original image.*
*2. Negative: Please remove the person's clothing, located at the upper right part of the segmentation image.*
*- Labels:*

*1. Positive: leash.*
*2. Negative: person's clothing.*

## B ADDITIONAL RESULTS

**Benchmarks on COD10K**   Results in Table 6 show that CoT-Seg displays SOTA or competitive results regarding segmenting highly difficult camouflaged objects, where CoT reasoning will make the task easier.

Table 6: Quantitative evaluation on camouflaged images in *COD-10K* (Fan et al., 2020). Note that the models are blind to both dataset divisions.

| Method | Training free | Train split | | | | Test split | | | |
|---|---|---|---|---|---|---|---|---|---|
| | | Implicit query | | Explicit query | | Implicit query | | Explicit query | |
| | | gIoU | cIoU | gIoU | cIoU | gIoU | cIoU | gIoU | cIoU |
| LISA-13B-Llama2 Lai et al. (2024) | No | 65.0 | 60.3 | 66.5 | 59.7 | 63.4 | 55.8 | 66.5 | 58.0 |
| Vision-Reasoner-7B Liu et al. (2025b) | No | 77.1 | 77.1 | **77.6** | 76.6 | 76.9 | 76.3 | 77.4 | **75.0** |
| CoT-Seg | Yes | 77.9 | **78.8** | 77.5 | **78.0** | 77.7 | 77.9 | 77.5 | 74.9 |
| CoT-Seg with self-correction | Yes | **78.0** | 78.4 | 77.4 | 76.6 | **78.2** | **78.6** | **77.6** | 74.9 |

**Self-Correction Visual Examples**   We show additional self-correction examples in Figure 14.

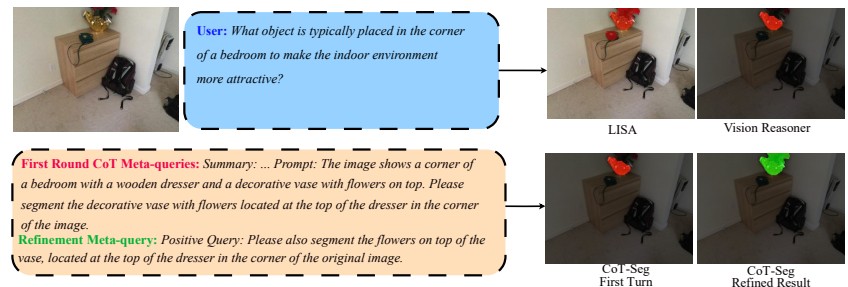

Figure 14: CoT-Seg can correct minor mistakes such as not segmenting the flowers with the vase.

## C SIMILAR WORKS ANALYSIS AND COMPARISON

**Vision Reasoner**   We discuss the difference between our work and the concurrent work Vision-Reasoner Liu et al. (2025b). To the best of our knowledge VisionReasoner uses reinforcement learning to generate the bounding boxes and segmentations. VisionReasoner has greatly improved on previous reasoning segmentation models as show in Tables 1– 6 but still fails in some complicated cases where there are a large number of objects to be segmented Fig 1–Fig 3 or when the prompt is very implicit. CoT-Seg in comparison, is zero-shot and can be easily plugged in to different models, offering high flexibility and achieves slightly higer scores in all the test data in Tables 1– 6.

**GSVA**   GSVA Xia et al. (2024) also uses MLLM to guide segmentation. Specifically, GSVA uses MLLM to generate [SEG] tokens and prompt the segmentation model towards supporting multiple object segmentation and a [NULL] token to reject absent object. In comparison, our approach uses chain of thought to provide information to the segmentator agent, empowering our model to solve very implicit queries and achieve multiple-object segmentation in a training-free manner. Our auto-correction process further leverages MLLM to improve and obtain accurate segmentations that the segmentor agent cannot achieve on its own. In RefCOCO tests in Table 1, GSVA achieves slightly higher results, mainly because of training and finetuning on the RefCOCO training dataset getting higher accuracy in prompts containing numerical postional arguments. e.g., Figure 12a and b. Further and similar training and finetuning of our model should be able to improve our results on these benchmarks, as well as incorporating our framework as a plugin to GSVA and other recent SOTA reasoning segmentation agents.

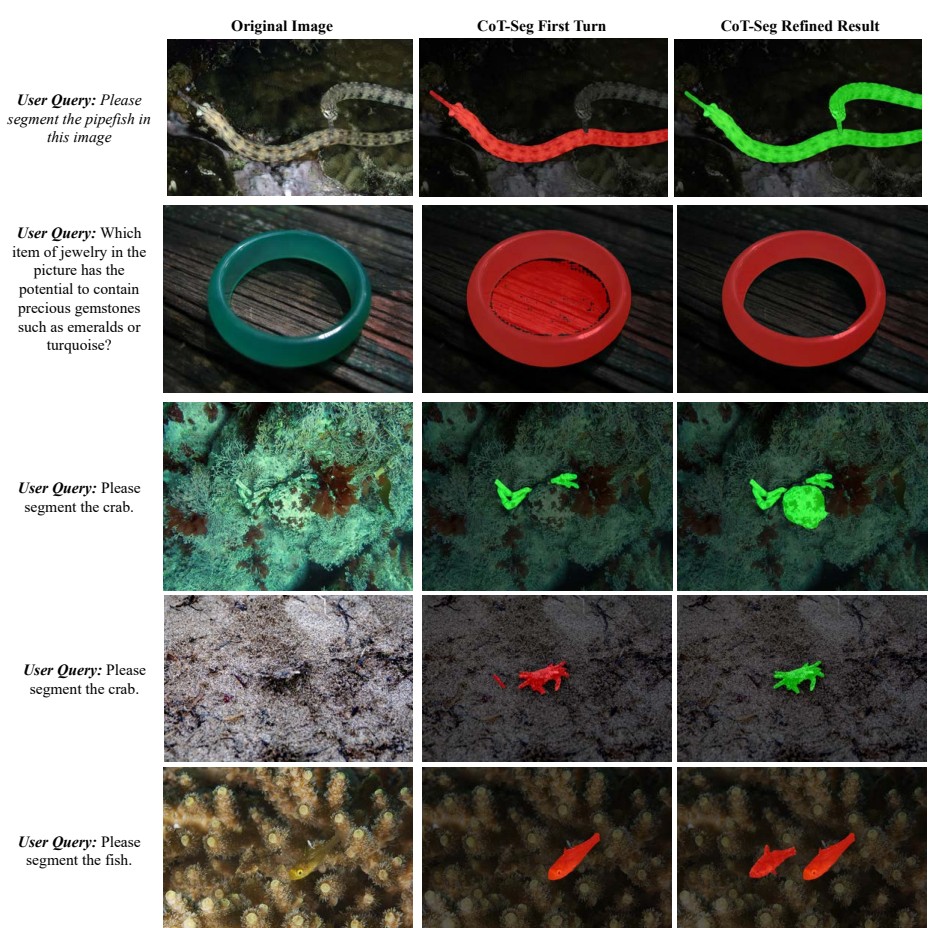

Figure 15: Additional self-correction results.

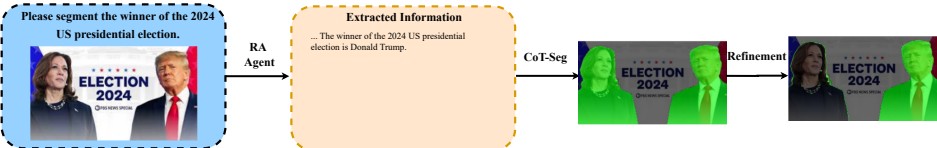

Figure 16: Retrieval-augmented CoT-Seg Result.

# D AUTO-CORRECTION

For most images in existing datasets auto-correction is not needed. In RefCoco and ReasonSeg only around 10% of the test cases required auto-correction, this is mostly because the segmentation task is usually quite simple and the segmentation agent is able to get it correct the first turn. Auto-correction usually happens in difficult cases when there is closely connected objects, or multiple objects similar to the object of interest. Thus, we believe in the future a more challenging dataset is beneficial, including difficult scenarios shown in Figures 1—4, to validate future stronger models that incorporate the ideas introduced by CoT-Seg in this paper.

# E LLM USAGE

Large Language Models (LLMs) were only used to polish the English sparsely in the paper.