# OpenReview forum: "CoT-Seg: Rethinking Reasoning Segmentation with Chain of Thoughts and Auto Correction"
_ICLR.cc/2026/Conference — ICLR 2026 Conference Withdrawn Submission_

### Official Review · Reviewer_aHGQ · 2025-10-16

**Soundness:** 2
**Presentation:** 2
**Contribution:** 2
**Rating:** 2
**Confidence:** 5

**Summary:**

This paper proposes CoT-Seg, a training-free framework that integrates chain-of-thought reasoning and self-correction for reasoning segmentation. The method decomposes input queries to effectively extract target information from implicit and complex prompts. Moreover, through the self-correction stage, the model evaluates and iteratively refines segmentation masks by checking the consistency between the original query and the generated reasoning trace. The proposed approach achieves strong performance on both ReasonSeg and RefCOCO datasets without requiring additional training.

**Strengths:**

- The paper proposes CoT-Seg, a training-free reasoning segmentation framework that integrates chain-of-thought reasoning with self-correction.


- The Reasoner leverages step-by-step thinking ability to address ambiguous prompt issues, while the Evaluator iteratively examines the consistency between the original prompt and the reasoning trace to refine the segmentation mask, thereby improving overall mask quality.

**Weaknesses:**

- The performance of CoT-Seg heavily depends on the models used within its components.
- As shown in Table 4, the performance of CoT-Seg varies depending on the segmentation backbone used for the Reasoner. Notably, when combined with LISA, the performance even decreases compared to using LISA alone, which raises concerns about the robustness and general applicability of CoT-Seg.
- In Section 4.5 (Ablation Studies), all experiments are conducted only on the RefCOCO Test-A set, which consists primarily of explicit queries. To better demonstrate the advantages of CoT-Seg, particularly its thinking ability and self-correction capability, it would be more appropriate to evaluate the method on ReasonSeg, which contains implicit and reasoning required queries.
- The authors claim that CoT-Seg can improve incorrect segmentation masks through self-correction. If so, it would be important to verify whether the framework can also reject empty-target queries, as done in GSVA. Following GSVA, evaluating with metrics such as N-acc could strengthen the analysis.
- It is unclear whether Retrieval Augmentation is always applied to the Reasoner module or only in specific scenarios. Clarifying this implementation detail would improve reproducibility.
- Equations (4)–(7) lack sufficient explanation; the meanings of variables such as s_P, s_N, and s’ are not clearly defined.
- Typo: Line 262 and Line 469.

**Questions:**

Please refer to the weaknesses.

---

### Official Review · Reviewer_AAuc · 2025-10-26

**Soundness:** 3
**Presentation:** 3
**Contribution:** 2
**Rating:** 2
**Confidence:** 4

**Summary:**

This paper proposes CoT-Seg, a training-free framework for reasoning segmentation that combines chain-of-thought (CoT) reasoning and self-correction using pretrained multimodal large language models and segmentation agents. The system decomposes complex segmentation queries into meta-instructions, generates initial masks via existing segmentors, and iteratively refines results through self-evaluation. Experiments prove effectiveness of the proposed method.

**Strengths:**

1. The idea of test-time reasoning with self-correction for segmentation is conceptually appealing.
2. Training-free nature is attractive for practical deployment.
3. Clear and well-written paper with systematic organization.

**Weaknesses:**

1. The work mainly combines existing paradigms (CoT reasoning, SAM-based segmentation, and self-correction loops) in a straightforward manner. While the integration is interesting, it lacks a fundamentally new insight on how to improve the model capacility for reasoning-based seg.
2. The reported improvements in Table 1-2 are marginal and sometimes within variance. It's hard to provide sufficient evidance of the effectiveness.
3. The proposed method combines several modules for segmentation without efficiency analysis. It could bring large computational overhead for incremental improvement.
4. There lacks a core comparison to evaluate the effectiveness of CoT pattern: Baseline W/O CoT vs. CoT-Seg.

**Questions:**

There are some suggestions:
1. Provide clearer analysis isolating the contribution of CoT reasoning versus baseline performance.
2. Include quantitative results showing how much self-correction alone contributes.
3. Explore open-source MLLM baselines for reproducibility.
4. Clarify failure cases and computational cost.

---

### Official Review · Reviewer_QgSM · 2025-10-27

**Soundness:** 2
**Presentation:** 3
**Contribution:** 2
**Rating:** 4
**Confidence:** 5

**Summary:**

Reasoning segmentation demands both accurate mask prediction and a sophisticated understanding of joint visual and linguistic cues. This paper introduces a training-free framework designed to enhance reasoning segmentation by incorporating chain-of-thought planning into the segmentation process. Complex language instructions are first decomposed by a multimodal large language model (MLLM) into structured, segmentation-ready directives. A self-correction mechanism then iteratively refines the segmentation output by feeding intermediate results back into the MLLM to mitigate false positives and false negatives. Furthermore, a retrieval-augmented reasoning module enables the system to consult external knowledge sources when the input provides insufficient contextual information. Experiments on ReasonSeg and RefCOCO tasks validate the effectiveness of the proposed pipeline, demonstrating consistent performance improvements.

**Strengths:**

The reasoning segmentation task requires a deep understanding of the vision-language command; therefore, it makes sense to leverage the MLLM as both the planner and evaluator, which provides both reasoning and self-correction abilities. The training-free manner also prevents the MLLM system or the segmentation system from overfitting to a specific dataset. Quantitative experiments on the reasonseg dataset show the improvement of the proposed pipeline. Ablation studies have been provided for the individual components, such as the CoT length, the segmentor, and MLLM.

**Weaknesses:**

The primary concern with this work lies in the limited experimental scope, both in terms of methodological details and benchmark coverage. Given that the proposed system is positioned as a training-free paradigm, it is particularly important to demonstrate its generalization across multiple datasets. However, the current evaluations appear quite restricted.

The authors state that “our method is training-free, we emphasize zero-shot evaluation to highlight the effectiveness of inference-time reasoning and self-correction.” This claim is not entirely accurate. The system leverages VisionReasoner as the SegAgent, which has already been fine-tuned on RefCOCO and ReasonSeg. As a result, it becomes difficult to disentangle where the observed reasoning capability originates—whether from VisionReasoner itself or from the proposed Seg-CoT pipeline. To more convincingly establish zero-shot generalization, the authors should evaluate the proposed pipeline with a VisionReasoner model not fine-tuned on LISA++ and report whether the relative improvements on ResonSeg differ from those in Table 2. Additionally, Table 4 suggests that performance drops considerably when replacing LISA or VisionReasoner-7B with Grounded-SAM, further indicating that the language-following ability primarily derives from the segmentation agent rather than the CoT reasoning module.

Regarding benchmark completeness: prior works routinely report results on RefCOCO, RefCOCO+, and RefCOCOg, capturing varied reasoning challenges, including fine-grained appearance cues, spatial relationships, and longer contextual descriptions. Instead, this paper focuses solely on RefCOCO, which largely consists of short expressions and predominantly single-target scenes. The authors are strongly encouraged to include gRefCOCO, which supports multi-target and no-target queries and therefore provides a more realistic and generalized evaluation. On the RefCOCO benchmark, the omission of VisionReasoner-7B baseline results is also problematic, since the model should be capable of producing segmentation directly without the proposed pipeline. Moreover, only cIoU scores are reported, even though cIoU is biased toward large objects; gIoU should be included for a fairer comparison. The authors provide an argument for why CoT-Seg does not outperform the baseline on RefCOCO, yet the ablation on CoT length shows only marginal gains (<0.5 gIoU), making the explanation less compelling. If RefCOCO is not a suitable benchmark for reasoning segmentation, then key ablations—such as CoT depth—should instead be conducted on ReasonSeg, where reasoning complexity is more prominent. Similarly, the improvements attributed to self-correction appear minor on both datasets, and the paper would benefit from additional analysis.



The paper also lacks ablations on meta-query format, even though the text explicitly acknowledges that the Segmentation Agent may accept different forms of prompts (e.g., text, box, points). Evaluating how the proposed reasoning framework interacts with these input modes would strengthen the claims of general applicability.

Finally, the manuscript does not include any computational efficiency analysis. Given that chain-of-thought reasoning and iterative self-correction naturally introduce inference-time overhead, a runtime comparison with baseline methods is essential to assess the practical trade-offs of the proposed approach.

**Questions:**

See the above

---

### Official Review · Reviewer_eDCA · 2025-11-01

**Soundness:** 3
**Presentation:** 2
**Contribution:** 1
**Rating:** 2
**Confidence:** 4

**Summary:**

The paper proposes CoT-Seg, a training-free framework composed of three stages to improve reasoning segmentation performance. In the first stage, given an image and a user query, the Reasoner agent uses Chain-of-Thought (CoT) techniques to produce question–answer pairs. After generating these Q–A pairs, the Reasoner agent summarizes them into a JSON-formatted meta-query compatible with the Segmentation Agent. Using this meta-query, the Segmentation Agent generates a segmentation mask. Then the Evaluator agent performs CoT reasoning to assess the mask and produces meta-queries for removing false positives and adding false negatives, which the Segmentation Agent uses to refine the mask. Additionally, when the image or query lacks sufficient information, a Retrieval Agent fetches relevant knowledge from a database (or the web) and supplies it to the Reasoner in the first stage. The authors claim that, on RefCOCO and ReasonSeg, their framework achieves comparable or superior performance.

**Strengths:**

1. The paper composes a carefully structured pipeline by combining multiple agents.
2. It provides many qualitative results, which aid understanding.

**Weaknesses:**

1. While the method improves reasoning for segmentation via a multi-agent setup, it risks error accumulation across agents (the paper itself acknowledges this (e.g., line 419). Such compounding errors depend heavily on the MLLM’s capability).
2. Achieving the level of qualitative performance shown appears to require costly models like GPT-4o, which challenges the claim that this is a general method. This is evident in Table 5.
3. Fundamentally, the approach seems incremental: it assembles existing models and techniques rather than introducing clearly novel algorithmic ideas specific to this framework. The shortcomings introduced by the multi-agent design are largely covered by using stronger models, not solved by the framework’s own innovations.
4. Moreover, given the added stages, the performance improvement does not seem compelling enough.

**Questions:**

1. For each qualitative result (Figs. 1–10), please specify the dataset used for the model comparisons in the captions.

2. How is the decision to trigger retrieval made? Does the Reasoner agent decide this?

3. Please expand the explanations of Equations (4)–(7) in Section 3.3.

4. In Tables 1 and 2, Most of other methods use 7B or 13B LLMs, whereas the paper compares against a setup using GPT-4o. Is this a fair comparison? Shouldn’t the comparisons be against Gemma 3 and Qwen2.5-VL-7B as in Table 5? Please reconstruct Tables 1 and 2 using those backbones for fairness.

5. In Table 3, why is the ablation conducted only on only RefCOCO? If CoT is introduced to handle reasoning-heavy tasks, shouldn’t the ablation be done on ReasonSeg, which requires more complex reasoning? Likewise, please add ablations using the models discussed in Q4.

6. Is the number of self-correction rounds a hyperparameter? It appears important so please include a corresponding ablation study.

7. In Section 4.5, line 439, isn’t the claim that fixed length is worse than variable length supported only by cIoU and not by gIoU? What criterion led to that conclusion?

8. A multi-agent pipeline likely increases inference time substantially. Compared with other methods, by how much does the average inference time increase? Please include a table.

---

### Note · Authors · 2025-11-13

I have read and agree with the venue's withdrawal policy on behalf of myself and my co-authors.